# Visualizing Changes in Global Glacier Surface Mass Balances before and after 1990

**Roger J. Braithwaite ***[ID] and Philip D. Hughes [ID]

Department of Geography, School of Environment, Education & Development University of Manchester, Manchester M13 9PL, UK; philip.hughes@manchester.ac.uk
* Correspondence: roger.braithwaite@manchester.ac.uk

**Abstract:** Recent satellite measurements of glacier mass balances show mountain glaciers all over the world had generally negative mass balances in the first decades of the 21st century. Mean summer temperatures all over the world rose from the 1961–1990 period to the 1991–2020 period, implying increasingly negative mass balances. We studied archived annual balances for 38 northern hemisphere glaciers to assess changes within the 1961–2020 period. We used a modified double-mass curve to visualize mass balance changes occurring around 1990. Mean balances in 1961–1990 were already small negative for many of the studied glaciers and became even more negative in 1991–2020 for glaciers in the Alps, at high latitudes and in western North America. The largest mass balance changes were for some glaciers in the Alps. We are unable to explain the lack of change in mean balance for one glacier in High Mountain Asia. We found complex changes for eight glaciers in Scandinavia, even including one glacier with a positive balance. We explain these changes by visualizing the deviations in winter and summer balances from their respective 1961–1990 mean values. High winter balances in the 1990s for Scandinavia partly obscured the emerging trend of increasingly negative summer balances, which we expect to continue in the future.

**Keywords:** glaciers; mass balance; summer temperature; annual balance; winter balance; summer balance; climate; climate change; double-mass curve

## 1. Introduction

Mountain glaciers all over the world had generally negative mass balances in the first decades of the 21st century [1–6]. This conclusion is mainly based on results from satellite altimetry and/or gravimetry. Rising temperatures and the resulting increases in glacier melting [7,8] are the likely causes of these negative balances. For example, the 1991–2020 mean summer temperature (June–August in the northern hemisphere) was warmer than the 1961–1990 mean summer temperature (Figure 1). The largest warming (1–2 °C) was in central Europe, including the Alps, the Mediterranean and in the Middle East, while temperature rises were 0–1 °C in other areas where glaciers occur. Glacier melting depends on summer temperatures [9,10], so Figure 1 supports the assertion that the negative mass balances of the early 21st century are due to rising temperatures.

Concerns about possible increases in ice melting in Greenland and on mountain glaciers stimulated the development of models that could simulate increased melting before it happened or, at least, before it was clearly visible (see references in [11,12]). To set the stage for the present paper, we quote some results from [13], who applied a degree-day model to median glacier elevations in seven different regions. For an assumed degree-day factor of 7 mm w.e. $d^{-1}$ $°C^{-1}$ for melting ice, they found average temperature sensitivities of annual balances of −0.81 to −1.00 (for Scandinavia) and −0.87 m w.e. $°C^{-1}$ (for the Alps).

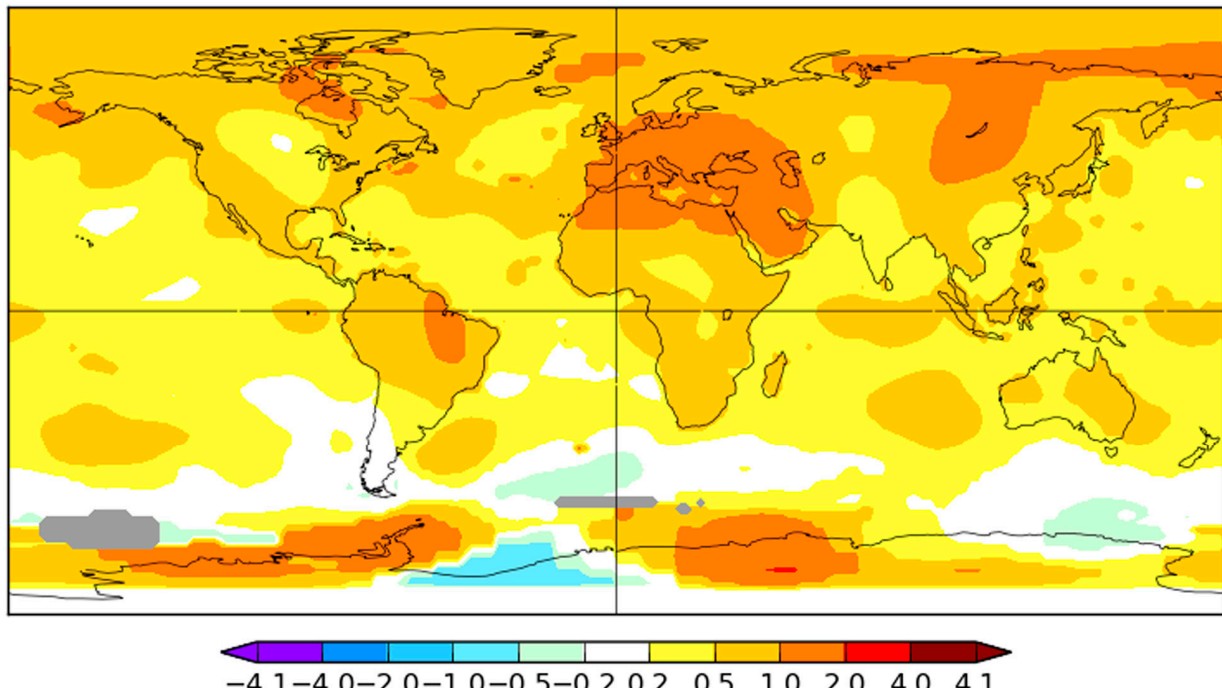

**Figure 1.** Global distribution of June–August temperature anomaly for 1991–2020 with respect to 1961–1990 base period. Adapted from [11,12]. Figure used here following the referencing guidelines: https://data.giss.nasa.gov/gistemp/references.html (accessed on 6 March 2024).

Aside from rising global sea levels, increased glacier melting will have profound effects on societies and environments close to the glaciers. It is therefore important to monitor ongoing glacier mass balance changes in different parts of the world and to update earlier predictions of what might still happen in the future. These predictions should be published in scientific papers and communicated to policymakers at international, national, and regional levels of government.

It is difficult to directly compare early 21st century measurements from satellites with late 20th century glacier mass balances because of the lack of global satellite coverage in the earlier period [1,2]. During the 20th century, surface mass balances were only available from arduous fieldwork on a few hundred individual glaciers [14–16] and most measured series were short and confined to only a few geographical regions.

The present paper identifies changes that must have occurred in recent decades, e.g., since about 1990, to turn mass balances from relatively close to zero for 1961–1990 [14] to the highly negative balances of the early 21st century detected by the satellites [4,5]. However, we can only achieve this for the few glaciers with more-or-less complete mass balance records over the past six decades [12]. The word 'global' in our title means that we looked through mass balance data from all over the world to find the few data that we could use for our purpose.

## 2. Materials and Methods

### 2.1. Observed Glacier Surface Mass Balances

In the present paper we discuss the data regarding glacier surface mass balances in Braithwaite and Hughes [10], which include an extensive and critical review of such data and we do not need to repeat this here, although a few comments are needed.

Surface balances are measured at many points on the glacier using a network of stakes and snowpits, and the mass balance of the whole glacier is an average of these single-stake data. Such measurements require fieldwork in hostile and remote environments,

and workers face many hazards. Surface balance measurements are only available from a few hundred glaciers. See Figure 1 in [10]. As an alternative to the laborious field work, mass balances can now be measured from satellites [17,18]. For example, average balances for periods in the early 21st century were determined by satellite altimetry [5] and by the Gravity Recovery and Climate Experiment (GRACE) [4]. These data are multi-year averages and are invaluable for monitoring glacier mass balances on a global scale, but such data cannot resolve inter-annual and seasonal changes, which are needed to understand changes in mass balances.

We can trace the development of mass balance concepts and methods through [15–27]. The oldest ongoing series of surface balance measurements were started in 1946 on Storglaciären in northern Sweden [21] and similar programs soon started elsewhere, partly guided by the International Hydrological Decade Program in 1965–1974 [27].

The database of the World Glacier Monitoring Service (WGMS) [28] lists all known data for glacier surface mass balances. We encourage readers to browse the WGMS database [28] to check that their data are included. Figure 1 in [10] shows the length of mass balance records up to 2020 from 482 glaciers in the WGMS database [28]. Two very long records are for Swiss rescue data [29,30], but most of the records are very short. For example, the mean length of the series is 13.6 years, and the median length is only 6 years. This shows that most mass balance studies run only for a few years on any individual glacier. The available data, therefore, may not well represent the full spectrum of glacier mass balance conditions in the world.

Seasonal balances are available for some of the above glaciers, and we can write the mass balance equation [17] as

$$B_{a,t} = B_{w,t} + B_{s,t} \tag{1}$$

where $B_{a,t}$ is the annual balance in the year $t$, $B_{w,t}$ is the winter balance, and $B_{s,t}$ is the summer balance, all in units of m w.e. a$^{-1}$. The latter balances are applicable if there is a definite winter season when most snow accumulation occurs, so the winter balance relates to the maximum transient balance at the end of the winter season [24]. Glaciologists do not always measure winter and summer balances, even when applicable, for reasons of logistics and/or economy.

Table 1 in [10] notes the greatly expanded network for 1961–2020, with more than half the available records (58%) in the last three decades (1991–2020). For the present study, therefore, we compare mass balances for the past two 30-year periods.

### 2.2. Mass Balance Changes from 1961–1990 to 1991–2020

The change in annual balance between periods 1 and 2 is

$$\Delta B_{a,2-1} = \overline{B}_{a,2} - \overline{B}_{a,1} \tag{2}$$

where the overbar denotes time-averages of annual balances $\overline{B}_{a,2}$ and $\overline{B}_{a,1}$ for periods 1 and 2. The mean annual balance in period 2 $\overline{B}_{a,2}$, according to Equations (1) and (2), results from changes in both winter $\Delta B_{w,2-1}$ and summer $\Delta B_{s,1-2}$ balances from period 1 to 2 according to

$$\overline{B}_{a,2} = \overline{B}_{a,1} + \Delta B_{w,1-2} + \Delta B_{s,1-2} + error \tag{3}$$

Although Equations (1) and (2) are exact, the error term in (3) is needed if winter and summer balances are available for slightly fewer years than annual balances. These errors are small for the glaciers considered in this study [10]. Equation (3) describes a conservation law for mass balance which shows that mean balances only change between periods when there are changes in winter and/or summer balances. You might measure mean balance $\overline{B}_{a,2}$ for some glaciers for a period in the early 21st century, but you cannot explain the result you obtain without knowing the other terms in Equation (3).

For this study, we chose periods 1 and 2 to be 1961–1990 and 1991–2020, respectively, to fit the definition of climate reference periods from the World Meteorological Organization (WMO) [31]. We therefore needed nearly complete mass balance records for each period.

Figure 2 shows locations of the glaciers used in this study with data from the WGMS database [28]. The glaciers do not include any from Greenland, South America, Iceland, east Africa, southern Asia, or New Zealand because of the shortness of records in those regions. Appendix A Table A1 lists the coordinates and periods of records for the chosen glaciers. This includes some updates to the data used for [10].

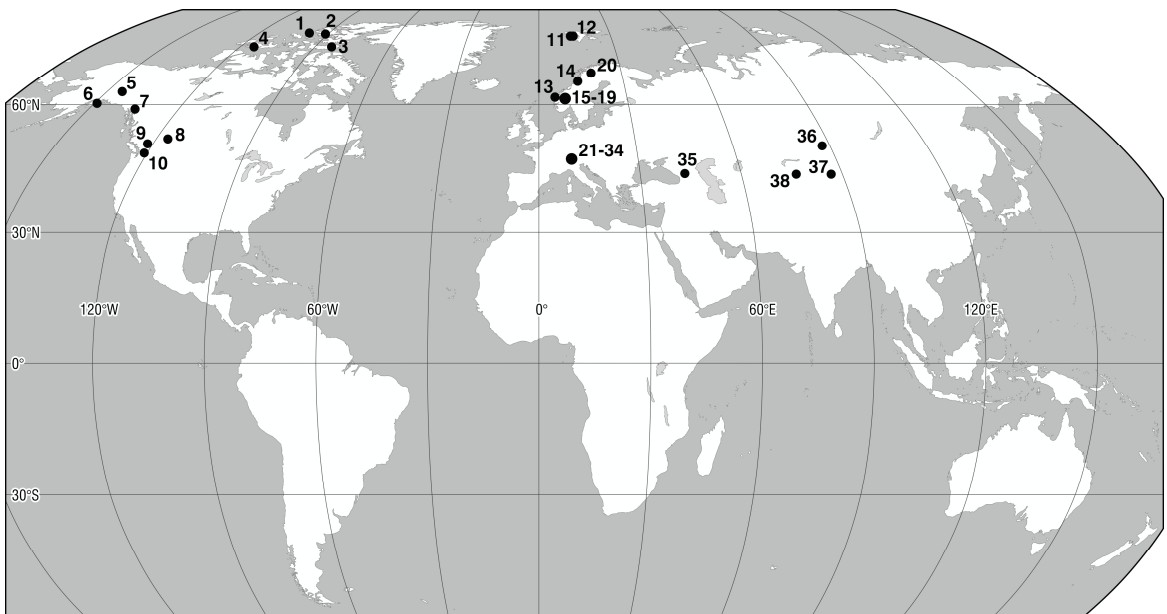

**Figure 2.** Locations of glaciers with nearly complete mass balance records for 1961–2020. Data from World Glacier Monitoring Service (WGMS) database [28].

Most of the series cover at least the full period 1961–2020, but some start after 1960/61. We allowed a few years of 'missing data' at the start of records as there seemed to be no strong trends in the 1961–1990 period, but we avoided missing data at the end of 1991–2020 because of the strong trends in that period, applying to almost all glaciers. The graphical approach outlined below readily takes account of missing data at the start of records better than a list of numbers as in [10].

Mass balance data typically show large inter-annual variations [32] and we needed to detect changes in mean balances against this background noise. Hydrologists and climatologists have long used double-mass curves [33] to detect relative changes in mean values. A double-mass curve involves plotting cumulative sums of two variables against each other. Following Tangborn [34], we can plot cumulative mass balances against calendar years if we regard the latter as the cumulative sum of a time variable.

There are hundreds of cumulative balance plots in the published glacier literature (for example, Figure 1 in [35]), but Braithwaite and Hughes [9] introduced an important modification that we use here. Instead of starting each cumulative curve from zero in the year before the record starts, as in the conventional approach, we shifted cumulative balance curves, so they were all zero for the year 1990. We chose 1990 so we could compare mass balances in the 1961–1990 and 1991–2020 periods, but other researchers can choose other years to suit their purposes.

## 3. Results

### 3.1. Using Cumulative Balance Plots to Visualize Mass Balance Changes

We show some cumulative balance plots in Figure 3a to illustrate the approach. These plots are based on some simple models that we devised to train ourselves in the interpretation of cumulative balance plots. Three model scenarios have small balances (negative, constant, and positive) for 1961–1990, followed by linear trends of increasingly negative balances in 1991–2020. A downward-trending (upward-trending) segment in Figure 3a

indicates an ever more negative (positive) mean balance, and a 'flat' segment indicates constant zero balance. The straight segments in 1961–1990 are for constant mean balances in the simple models, while the slightly concave curves in 1991–2020 reflect linear trends in mean balances. A step change in model mass balance (not shown here) from one constant mean value to another would give two straight lines meeting in the year of change. Aside from adjusting cumulative balance curves to pass through zero in 1990, we plotted curves in different graphs with the same scales on the *x-y* axes to help with visualization, even if it left large blank areas in some graphs.

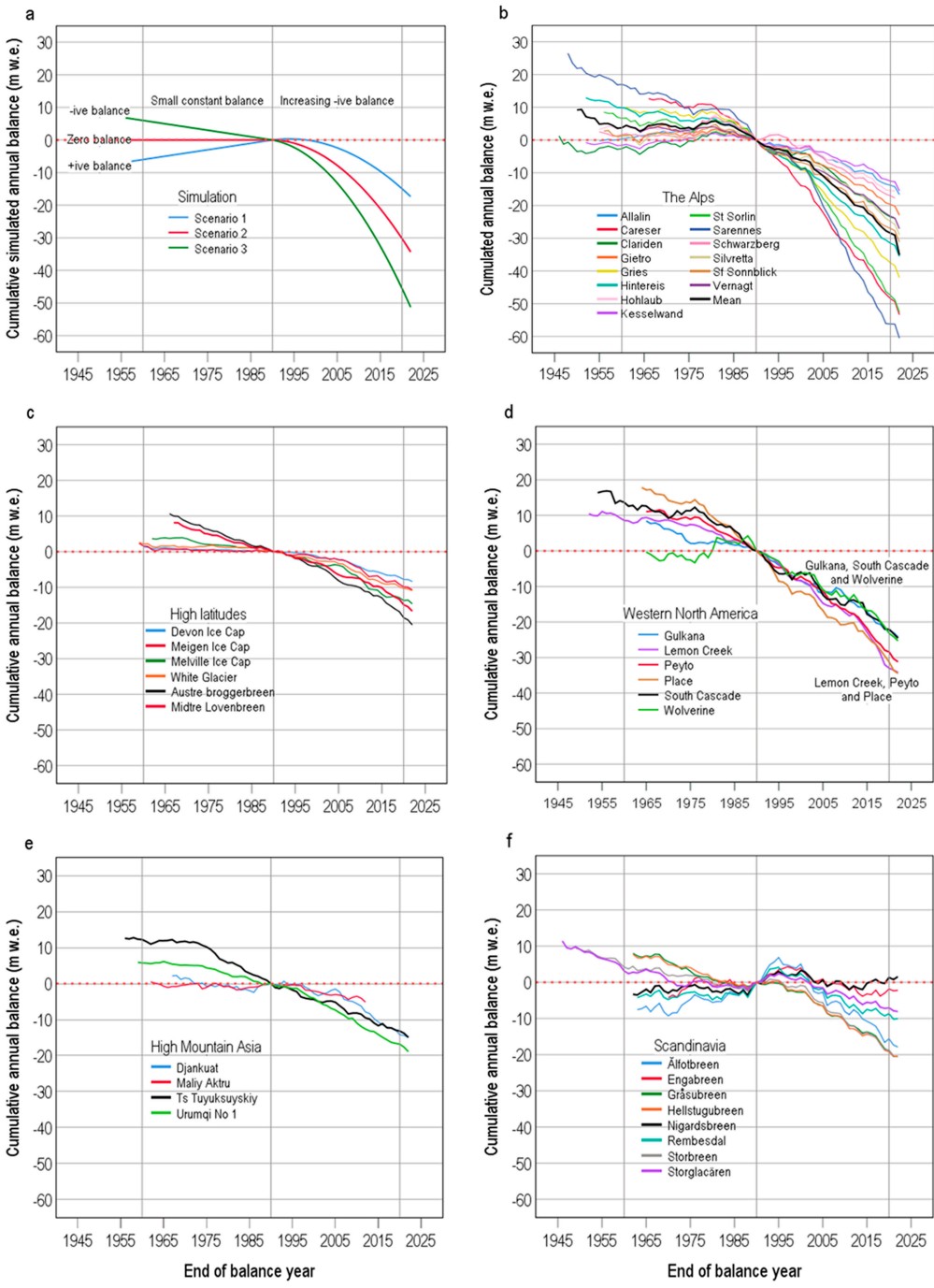

**Figure 3.** Cumulative balance plots for some model simulations (**a**), and for annual balances in the Alps (**b**), at high latitudes (**c**), in western North America (**d**), in High Mountain Asia (**e**) and in Scandinavia (**f**). See Section 3.1 for modified double-mass curve. All annual balance data are from the WGMS database [28].

With a cumulative balance equal to zero in 1990 in Figure 3a, the cumulative balance in 2020 is equal to the mean balance for the period 1991–2020 multiplied by 30 years of the record in the period. If the cumulative balance was zero in 1960, the cumulative balance in 1990 would be the mean balance for 1961–1990 multiplied by 30 years. However, shifting of the cumulative balance curve to pass through zero in 1990 means that the cumulative balance in 1960 is equal to the mean balance for 1961–1990 multiplied by −30. With this interpretation, we can visualize changes in the mean mass balance from one period to the next by comparing cumulative balances in 1960 and 2020, respectively. For the three scenarios in Figure 3a, we can say that simulated mass balances changed from small (negative, zero or positive) in the first period to strongly negative in the second period.

*3.2. Mass Balance Changes in Five Regions*

We plotted cumulative balances in Figure 3b for 14 glaciers in the Alps (glaciers numbers 21 to 34 in Figure 2). In the case of missing data at the start of records we can easily estimate where the curves would intersect the 1960 line. From the values of the cumulative balances in 1960 and 2020, respectively, we visualized mean balances changing from moderately negative or small positive in 1961–1990 to strongly negative in 1991–2020. The mildly concave curvature of the cumulative balance plots for 1991–2020 shows a trend of increasing mean balances after the balance year 1979/80, which is consistent with the change in summer temperatures reported by [9,10].

We started our study with Alpine glaciers (Figure 3b) as they seem to change in a relatively simple way, all trending to increasingly negative balances after about 1980, but at varying rates. In [9] we suggest the latter reflects differing temperature sensitivity to a similar temperature variation across the whole Alpine glacier cryosphere. We can now visually assess mass balance changes in other regions (Figure 3c–f) to compare with those for Alpine glaciers (Figure 3b).

Figure 3c shows cumulative balances for six glaciers at high latitudes (glaciers 1 to 4 and glaciers 11 and 12 in Figure 2). Although full data are only available for one glacier for 1960, we can see that mean balances changed from small negative in 1961–1990 to moderately negative for 1991–2020. This is a much smaller change than for Alpine glaciers (Figure 3b), but consistent with models suggesting smaller temperature sensitivity for high latitude glaciers [13], combined with smaller summer temperature changes at high latitudes than for Alpine glaciers [12].

Figure 3d shows cumulative balances for six glaciers in western North America (glacier numbers 5 to 10 in Figure 2). Two of the glaciers cover the full period of 1961–2020, while four series started after 1960. Mean balances change from small or moderately negative for 1961–1990 to moderately or strongly negative in 1991–2020. The range of 2020 cumulative balances is narrower than for Alpine glaciers (Figure 3b), and changes in mass balances from 1961–1990 to 1991–2020 are generally smaller. Curiously, the 2020 cumulative balances for western North America fall into two groups: (1) Lemon Creek, Peyto and Place; (2) Gulkana, South Cascade and Wolverine. This grouping is difficult to explain in simple geographical terms and needs more research.

Figure 3e shows cumulative balances for four glaciers in High Mountain Asia (glacier numbers 35 to 38 in Figure 1). The visual impression from Figure 3e is of small changes of mean mass balance for three of the glaciers, suggesting low temperature sensitivity of mass balances. However, there is very little or no change in mean mass balance between the two periods for Ts. Tuyuksuyskiy, despite a rise in temperature [10]. We cannot explain this.

Figure 3f shows cumulative balances for eight glaciers in Scandinavia (glacier numbers 13 to 20 in Figure 2). Seven are from Norway and one from northern Sweden (Norwegian and Swedish data are seldom plotted on the same graph). Readers can compare Figure 3f with Figure 1.5 in [35] to see which format they prefer. Cumulative balances for 1960 are within the range small negative to small positive, and cumulative balances for 2020 are moderately negative for four of the glaciers, small negative for three glaciers and even

positive for one glacier. The latter is an astonishing result, and we discuss this 'Scandinavian anomaly' in the following section.

### 3.3. The 'Scandinavian Anomaly'

Figure 3f shows surprisingly small mass balance changes from 1961–1990 to 1991–2020 despite a rise of about 0.7 °C in summer temperature over the same period [10]. Modeling studies of mass balance variations generally agree that the annual balance is correlated with both summer temperature and some kind of precipitation variable [36], and we therefore try to explain this apparent anomaly by looking at variations in winter and summer balances. As winter and summer balances each have positive and negative magnitudes of several m w.e. a$^{-1}$, respectively, cumulative balances would plot off the scales used for Figure 3a–f. We avoid this by plotting deviations from the respective mean balances for 1961–1990. Equation (4) defines the winter balance deviation from the 1961–1990 mean for winter balance:

$$B_{w,t}^* = B_{w,t} - \overline{B}_{w,1961-90} \tag{4}$$

and we define deviations of annual and summer balances in a similar way. According to Equation (3), the mean balance for 1991–2020 depends on the mean balance for 1961–1990. The calculation of balance deviations removes this legacy effect of earlier mass balances.

We re-plotted cumulative annual balances for the eight Scandinavian glaciers in Figure 4a using a more appropriate scale for the vertical axis. We also marked the three most continental glaciers, Gråsubreen, Hellstugubreen and Storbreen according to [35] with dotted lines. With this modification, the mystery of the Scandinavian glaciers deepens. The three most continental glaciers have the greatest negative cumulative balances for 2020, but the most maritime glacier according to [35], Ålfotbreen, has the next most negative cumulative balance, and another maritime glacier, Nigardsbeen, even has a positive cumulative balance for 2020. This Scandinavian anomaly cannot be explained simply in terms of differences between maritime and continental glaciers.

We plotted cumulative balance deviations for the eight glaciers in Figure 4b. The difference between this plot and the previous one (Figure 4a) is due to the elimination of the legacy effect of 1961–1990 balances. The maritime glacier Ålfotbreen has a strongly negative cumulative balance deviation in 2020, followed by the continental glacier Storbreen. The other two continental glaciers have moderately negative cumulative balance deviations. A notable feature of the plots in Figure 4b is that cumulative balance deviations have positive trends in the early 1990s, which are followed by downward trends from the late 1990s. Ålfotbreen has the highest of the positive cumulative balance deviations in the early 1990s. If it were not for this, the cumulative balance deviation for Ålfotbreen in 2020 would be about 5 m w.e. more negative than it is.

Figure 4c,d show plots of cumulative winter and summer balance deviations, respectively. There are only small trends towards increased winter balances (Figure 4c) in 1991–2020, but there are high winter balance trends in the early 1990s due to a clustering of years with high winter balances. These continue for the whole period 1991–2020 for Ålfotbreen and Nigardsbeen, while the three continental glaciers show a small net decrease in winter balance by 2020. Figure 4d shows that summer balances for all Scandinavian glaciers became more negative from the late 1990s onwards. The summer balance deviations in the early 1990s were probably positive due to the less efficient melting of increased snow cover compared with the melting of bare ice [37]. The increased snow cover can, in turn, be attributed to high positive indices for the North Atlantic Oscillation (NAO) or Arctic Oscillation (AO) [38].

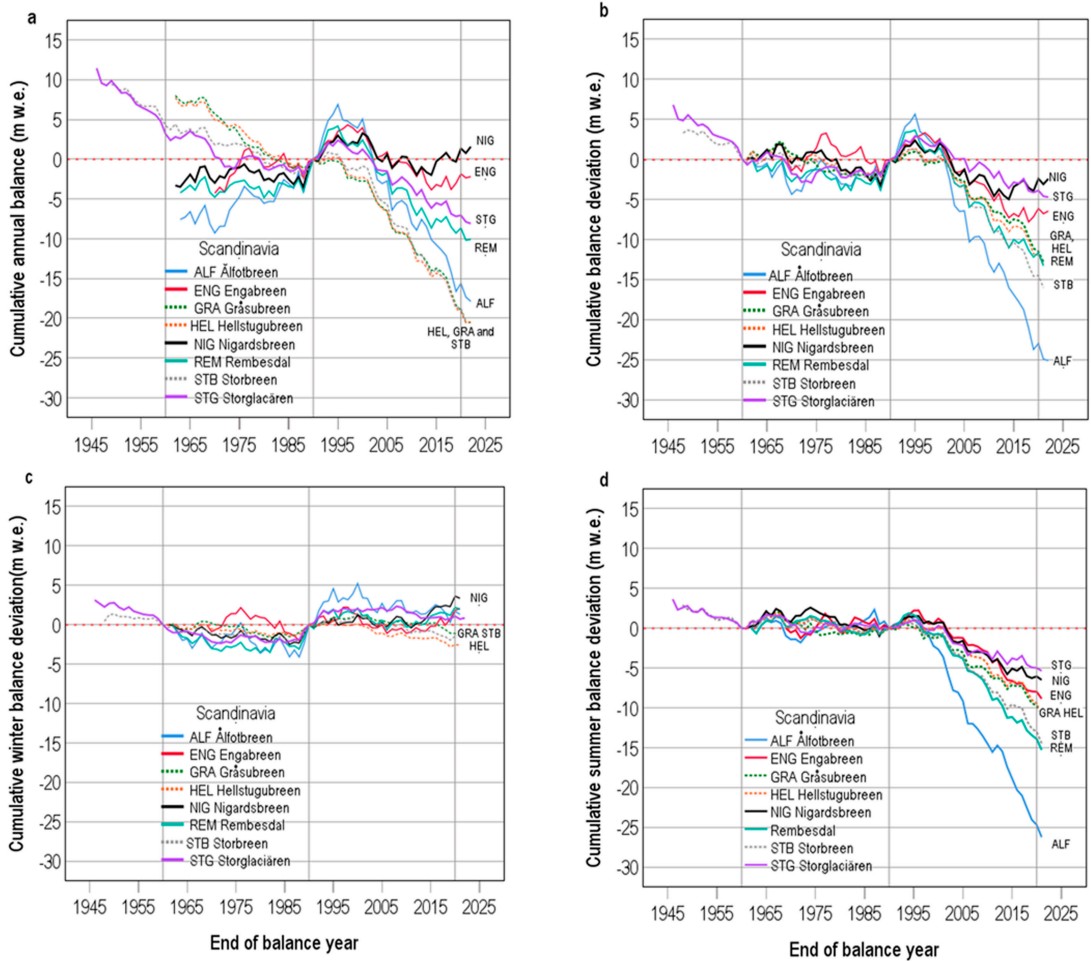

**Figure 4.** Cumulative plots for annual balance (**a**), balance deviation (**b**), winter balance deviation (**c**) and summer balance deviation (**d**) for eight glaciers in Scandinavia. See Section 3.1 for modified double-mass curve. All balance data are from the WGMS database [28].

Comparing summer balance deviations (Figure 4d) and winter balance deviations (Figure 4c) with annual balance deviations (Figure 4b) suggests that the latter are mainly explainable by variations in summer balances, modified to some extent by smaller variations in winter balances. The plots in Figure 4d are consistent with increased melting for all eight glaciers from the early 1990s onwards, which was partly offset by short-term increases in winter balances that were large enough to mask what were then small increases in melting. Further research is needed to completely explain the differing temperature sensitivities of these eight Scandinavian glaciers, but balance deviations for these glaciers will probably become increasingly negative as summer temperature continues to rise above the level of winter balance fluctuations.

## 4. Discussion

Figure 3b–f show two anomalies compared with Alpine glaciers. We cannot explain the apparent anomaly for Ts. Tuyuksuyskiy (Figure 3e), but there are several research groups active in the area who may be able to achieve this. For example, we only considered long series of measured surface balances to detect balance changes within the 1961–2020 period, and the sparsity of early mass balance measurements limits our approach to only a few glaciers. Better geographical coverage including glaciers of various types [39] is needed. Researchers on Asian glaciers might combine available short series of surface balance measurements with satellite measurements, and with glacier–climate modeling to detect 1961–2020 changes. For example, mass balance measurements started on some

glaciers in the former USSR around 1957 [40] and continued to about 1990. Combining these early mass balance data with recent satellite imagery may provide the desired changes for the six decades of 1961–2020.

The anomaly for Scandinavian glaciers (Figure 3f) probably results from variations in winter balances (Figure 3c) partly masking the effects of rising temperature on summer balances (Figure 3d). The winter balances of some Scandinavian glaciers are very high compared with other glaciers in Table A1, and they also have very high year-to-year variations [32]. For example, the standard deviations of winter balances for maritime glaciers like Ålfotbreen, Engabreen and Nigardsbeen are about $\pm 0.5$ to $\pm 1.0$ m w.e. $a^{-1}$, while the standard deviations for glaciers in the Alps are only $\pm 0.2$ to $\pm 0.6$ m w.e. $a^{-1}$. The increasingly negative trends due to rising summer temperatures will be further reinforced if winter balances decrease substantially in the future.

Our modification of the double-mass curve for easy visualization of long-term variations in glacier mass balances is quite effective but is limited in usefulness by the rarity of long mass balance records. However, the approach can be modified to suit circumstances. For example, there are many mass balance records in western North America starting in the 1980s [41] and their 2020 cumulative balances (relative to zero in 1990) could be compared with the few cases covering the full period in Figure 2d. Similar modifications could be made for South American and Icelandic glaciers with shorter mass balance records.

Our emphasis on long records of mass balance data means that most of our data comes from Europe and North America. We encourage glaciologists in other areas to continue, or even expand, mass balance measurements for their own purposes and to serve their communities, and to develop methods more suited to short mass balance series.

## 5. Conclusions

The modification of the double-mass curve commonly used in studies of glacier mass balances is effective in visualizing changes in mass balances. Applying the approach to the few glaciers with nearly complete data for 1961–2020 shows that average annual balances generally changed from small (negative or positive) to increasingly negative after the 1980s or 1990s. One exception to this is in High Mountain Asia, which we cannot explain, but another apparent exception in Scandinavia (eight glaciers) is resolved by visualization of winter and summer balances rather than just annual balances.

**Author Contributions:** Conceptualization and methodology, R.J.B.; writing original draft, review and editing, R.J.B. and P.D.H.; visualization, R.J.B.; supervision and project administration, P.D.H. All authors have read and agreed to the published version of the manuscript.

**Funding:** This research received no external funding.

**Institutional Review Board Statement:** Not applicable.

**Informed Consent Statement:** Not applicable.

**Data Availability Statement:** The data presented in this study are available on request from the corresponding author. The data are not publicly available due to the privacy.

**Acknowledgments:** The University of Manchester has supported R.J.B. with an Honorary Senior Research Fellowship since 2010. We use archived glacier and climate data from various agencies including World Glacier Monitoring Service (WGMS), NASA/GISS and CRU/UEA and we are deeply grateful to all those who fund, compile, validate, and distribute such data for use by the wider community.

**Conflicts of Interest:** The authors declare no conflicts of interest. There were no funders.

## Appendix A

**Table A1.** Latitude and longitude of 38 glaciers with nearly complete annual balance data for 1961–2020. WGMS is the reference number in the World Glacier Monitoring System (WGMS) database [28]. Bw denotes availability of winter balance data for the glacier. Start and End denote start and end of the annual balance record.

| Glacier Name | WGMS | Country | Lat. | Long. | Bw | Start | End |
|---|---|---|---|---|---|---|---|
| | | | (° N) | (° E) | (Y/N) | (a) | (a) |
| Meighen Ice Cap | 16 | Canada | 79.9 | −99.1 | Y | 1960 | 2022 |
| White Glacier | 1 | Canada | 79.5 | −90.9 | N | 1960 | 2022 |
| Devon Ice Cap NW | 39 | Canada | 75.4 | −83.3 | Y | 1961 | 2022 |
| Melville Ice Cap | 3690 | Canada | 75.4 | −115.0 | N | 1963 | 2022 |
| Gulkana | 90 | USA | 63.3 | −145.4 | Y | 1966 | 2022 |
| Wolverine | 94 | USA | 60.4 | −148.9 | Y | 1966 | 202 |
| Lemon Creek | 3334 | USA | 58.4 | −134.3 | N | 1953 | 2022 |
| Peyto | 57 | Canada | 51.7 | −116.5 | Y | 1966 | 2022 |
| Place | 41 | Canada | 50.4 | −122.6 | Y | 1965 | 2022 |
| South Cascade | 205 | USA | 48.3 | −121.0 | Y | 1953 | 2022 |
| Aust. Brøggerbreen | 292 | Svalbard | 78.9 | 11.8 | Y | 1967 | 2022 |
| Midtre Løvenbreen | 291 | Svalbard | 78.9 | 12.0 | Y | 1968 | 2022 |
| Ålfotbreen | 317 | Norway | 61.8 | 5.7 | Y | 1963 | 2022 |
| Engabreen | 298 | Norway | 66.7 | 13.9 | Y | 1970 | 2021 |
| Gråsubreen | 299 | Norway | 61.7 | 6.6 | Y | 1962 | 2021 |
| Hellstugubreen | 300 | Norway | 61.6 | 8.4 | Y | 1962 | 2021 |
| Nigardsbreen | 290 | Norway | 61.7 | 7.1 | Y | 1962 | 2022 |
| Rembesdal | 2296 | Norway | 60.5 | 7.4 | Y | 1963 | 2022 |
| Storbreen | 302 | Norway | 61.6 | 8.1 | Y | 1949 | 2021 |
| Storglaciären | 332 | Sweden | 67.9 | 18.5 | Y | 1946 | 2022 |
| Allalin | 394 | Switzerland | 46.0 | 7.9 | Y | 1956 | 2022 |
| Careser | 635 | Italy | 46.5 | 10.7 | N | 1967 | 2022 |
| Clariden | 260 | Switzerland | 46.8 | 8.9 | Y | 1915 | 2021 |
| Gietro | 367 | Switzerland | 46.0 | 7.4 | Y | 1967 | 2022 |
| Gries | 359 | Switzerland | 46.4 | 8.3 | Y | 1962 | 2022 |
| Hintereis | 491 | Austria | 46.8 | 10.8 | N | 1953 | 2022 |
| Hohlaub | 3332 | Switzerland | 46.1 | 7.9 | Y | 1957 | 2021 |
| Kesselwand | 507 | Austria | 46.8 | 10.8 | N | 1953 | 2022 |
| Sarennes | 357 | France | 45.1 | 6.1 | Y | 1949 | 2022 |
| Schwarzberg | 395 | Switzerland | 46.0 | 7.9 | Y | 1956 | 2021 |
| Silvretta | 408 | Switzerland | 46.8 | 10.1 | Y | 1919 | 2022 |
| St. Sonnblick | 573 | Austria | 47.1 | 12.5 | N | 1957 | 2021 |
| Saint Sorlin | 356 | France | 45.2 | 6.2 | N | 1957 | 2022 |
| Vernagt | 489 | Austria | 46.9 | 10.8 | Y | 1965 | 2022 |
| Djankuat | 726 | Russia | 43.2 | 42.8 | Y | 1968 | 2022 |
| Maliy Aktru | 795 | Russia | 50.0 | 87.7 | N | 1962 | 2012 |
| Urumqi No. 1 | 853 | China | 43.1 | 86.8 | N | 1959 | 2022 |
| Ts. Tuyuksuyskiy | 817 | Kazakhstan | 43.0 | 77.1 | Y | 1957 | 2022 |

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
