# Peer review of "Visualizing Changes in Global Glacier Surface Mass Balances before and after 1990"

_atmosphere, doi:10.3390/atmos15030362_

Round 1

Reviewer 1 Report

Comments and Suggestions for Authors

The study of short-term glacierespecially on a ten-year scale mass balance is of great significance for water resources and future global climate change. This paper study archived annual balances for 38 Northern Hemisphere glaciers to assess 10 changes within in the 1961-2020 period. They used a modified double-mass curve to visualize mass 11 balance changes occurring around 1990. Research shows thay the modification of the double-mass curve commonly used in studies of glacier mass 330 balance is effective in visualizing changes in mass balance. The paper reveals the characteristics of glacier mass balance changes in different research areas, and explaines some abnormal phenomena in glacier mass balance. I am very interested in the research presented in this article. Moreover, the improvement method proposed in this article provides a good reference for analyzing the mass balance of glaciers in other regions. I encourage this article to be published.

The minor improvements and revisions in the text are, 1) Athough the Bw denotes availability of winter balance data for the glacier, the Y/N in Bw denotes availability of winter balance data for the glacier AppendixA need to be further clarified. 2) In AppendixA, the end year of the Wolverine should be 2022

Author Response

Reviewer no. 1

Thank you for your favourable review. You point out a couple of points for improvement. We thank you for this and will make those corrections.

The editor's office said we should reduce self-citations. We are looking into this.

Roger Braithwaite

Reviewer 2 Report

Comments and Suggestions for Authors

Glacier mass balance plays a crucial role in understanding the impacts of climate change on Earth's cryosphere. This manuscript presents an analysis of glacier mass balance changes across the Northern Hemisphere using a modified double-mass curve approach. Through examination of archived annual balances for 38 Northern Hemisphere glaciers spanning the period from 1961 to 2020, the authors reveal a general shift towards increasingly negative mass balances after the 1980s or 1990s, indicative of ongoing glacier retreat. Notably, exceptions in High Mountain Asia and Scandinavia challenge the overall trend, underscoring the complexity of regional glacier dynamics.

General comments

1. Figures

(1)  The inclusion of Figure 1 appears redundant as it closely resembles Figure 2 in Braithwaite and Hughes (2023). Given that Figure 2 in the referenced work adequately illustrates global temperature changes, Figure 1 in the manuscript provides no additional useful information. Readers can gain a comprehensive understanding of global temperature trends by referring to Braithwaite and Hughes (2023), rendering the presence of Figure 1 unnecessary.

(2)  Could you improve the quality Figures 3 and 4? It is not easy to see the curve changes of individual glaciers. Vector images maybe more appropriate here.

(3)  In Figure 3a, the words of 'positive' and 'negative' are incomplete with only the suffix '-tive' is given.

2. I think it would be more appropriate by put the first paragraphs in Section 3.1 into section 2 (Materials and Methods)

3. Could you please provide clarification on the criteria used to classify glaciers in Scandinavia as either continental or maritime? It would be helpful to reference relevant literature that elucidates the characteristics and distinctions between these glacier types.

Specific comments

lines 78-80: This long sentence should be more concise.

line 141: delete Fig 2 as Figure 2 is given.

line 143: This sentence seems a comment from coauthor or editor.

line 187: why you set the factor value to ±30?

line 189: you mentioned the simulated mass balance. How do you simulate and which work in the manuscript is about simulation?

line 238: This section should be 3.3.

line 245: Figure 3a to 3f ---> Figures 3a to 3f

lines 266-267: what is legacy effect?

line 292: a typo for two dot symbols

line 329: change "Conclusions" to "5. Conclusions"

Comments on the Quality of English Language

As a non-native English speaker, I found the overall quality of English language in the manuscript to be proficient.

Author Response

Reviewer no. 2

Thank you for your generally favourable review. Please see my answers in the uploaded document.

The editor's office said we should reduce self-citations. We are looking into this.

Roger Braithwaite

Reviewer 3 Report

Comments and Suggestions for Authors

The article is valuable. because it provides information about trends in the mass balance of glaciers in different regions of the Earth. Of greatest interest is the consideration of the Scandinavian mass balance anomaly.

Author Response

Reviewer no. 3

Thank you for your favourable review.

The editor's office said we should reduce self-citations. We are looking into this.

Roger Braithwaite